# HPV Vaccination and CIN3+ Among Women Aged 25–29 Years in Northern Norway, 2010–2024: A Population-Based Time-Series Analysis

**DOI:** 10.3390/vaccines13111147

**Published:** 2025-11-09

**Authors:** Sveinung Wergeland Sørbye, Mona Antonsen, Elin Synnøve Mortensen

**Affiliations:** 1Department of Clinical Pathology, University Hospital of North Norway, 9006 Tromsø, Norway; mona.antonsen@unn.no (M.A.); elin.mortensen@unn.no (E.S.M.); 2Department of Medical Biology, Faculty of Health Sciences, University of Tromsø, 9006 Tromsø, Norway

**Keywords:** HPV vaccination, cervical intraepithelial neoplasia, CIN3+, cervical cancer prevention, population-based study, time-series analysis, Norway, cervical cancer screening, HPV testing, vaccine impact, catch-up vaccination

## Abstract

Background/Objectives: Cervical intraepithelial neoplasia grade 3 and worse (CIN3+) is a robust surrogate for cervical cancer risk. In Norway, organized cervical screening starts at 25 years of age (25–69 years). Norway introduced school-based HPV vaccination with the quadrivalent vaccine for 12-year-old girls in 2009 (birth cohorts ≥ 1997) with high 3-dose completion, and a catch-up program with the bivalent vaccine for women born 1991–1996 in 2016–2019 with lower uptake. We assessed whether increasing birth-cohort vaccination coverage (defined as ≥1 dose) was followed by reductions in CIN3+ at the age of entry to organized screening (25–29 years). Methods: We conducted a retrospective, population-based time-series of women aged 25–29 years in Troms and Finnmark screened in 2010–2024. CIN3+ was counted per unique woman and expressed per 1000 screened women per year. Cohort-level vaccination exposure was proxied by birth-year eligibility and national coverage (≥1 dose) by calendar year. Temporal trends were assessed using segmented linear regression (2010–2017; 2017–2024). Results: Among 42,253 screening tests, 865 women had CIN3+. CIN3+ rates were stable in 2010–2016 (≈15–24 per 1000), peaked in 2017–2018 (≈26–28 per 1000), and declined to 6.6 per 1000 in 2024 (~75% reduction from the peak). The 2010–2017 trend was not significant (*p* = 0.244), whereas 2017–2024 showed a significant annual decline (slope −3.04 per 1000 per year; *p* = 7.4 × 10^−5^). The decline coincided with an increase in the vaccinated share of the age group from an estimated 12% in 2017 to 78% in 2024. Cervical cancer was rare throughout and absent in 2024, and the 2023 transition to primary HPV testing did not interrupt the downward trend. Conclusions: As vaccinated birth cohorts—especially those vaccinated before sexual debut—entered organized screening at age 25, CIN3+ in women aged 25–29 years fell markedly. Estimates are based on coverage defined as ≥1 dose; future linkage to individual dose data and HPV type–specific CIN3+ is warranted.

## 1. Introduction

Cervical cancer is caused by persistent infection with oncogenic human papillomavirus (HPV), most commonly HPV 16 and 18, and develops over years through histologically defined precursors (CIN2/3) [1]. Cytology-based screening was introduced to detect and treat these precancers before invasion [2], but cytology has only moderate sensitivity (≈50–70%), so some cancers occur despite previous normal smears [3]. HPV-based screening is more sensitive and provides substantially greater protection against invasive cervical cancer than cytology [4], but with lower specificity and a greater need for triage of HPV-positive women [5]. Against this background, cervical intraepithelial neoplasia grade 3 and worse (CIN3+) is a robust surrogate endpoint for cervical cancer risk and a key indicator for monitoring prevention strategies [6]. Prophylactic HPV vaccination has shown durable protection against vaccine-type high-grade disease: in the Nordic long-term follow-up of the FUTURE II trial, no HPV16/18-related CIN2+ was detected through 12–14 years of follow-up, corresponding to 100% effectiveness in women vaccinated as young adults [7]. Real-world data from Denmark have further documented reductions of about 49% in CIN2+ and 47% in CIN3+ at the population level after introduction of the quadrivalent vaccine, i.e., for lesions irrespective of HPV type [8]. Importantly, nationwide studies from Denmark and Sweden have shown that this prevention of high-grade precursors translates into lower invasive cervical cancer incidence: among women vaccinated before 17 years of age, cervical cancer was reduced by about 86% in Denmark and 88% in Sweden [9,10]. Together, these data support the use of CIN3+ as the most clinically consequential precancerous endpoint when assessing HPV vaccine impact.

Norway introduced HPV vaccination in 2009 for girls in the 7th grade (birth cohorts 1997 and later) using the quadrivalent vaccine (Gardasil) in a 3-dose schedule, with coverage (≥1 dose) increasing from about 70% in the earliest cohorts (2009–2011) to around 85% in intermediate cohorts and to over 90% in recent years [11,12]. Because completion of all three doses was high in the school-based program, ≥1 dose closely reflects full vaccination in these cohorts. Girls born in 1997 who were not vaccinated in 2009 were again offered HPV vaccination in the 2016–2019 national catch-up, further increasing coverage in this birth cohort. A national catch-up program in 2016–2019 offered HPV vaccination to women born in 1991–1996; this program used the bivalent vaccine (Cervarix) in a 3-dose schedule and achieved about 60% uptake of ≥1 dose, but dose completion was lower and many women were likely exposed to HPV before vaccination, which may attenuate impact compared with routine vaccination at age 12 [13]. In 2018, the childhood program changed from quadrivalent to bivalent vaccine and to a 2-dose schedule for girls < 15 years, but these younger cohorts had not yet reached the 25–29-year screening age and are therefore not included in the present analysis.

In Northern Norway (Troms and Finnmark), we have previously reported substantial declines in high-grade precancer following vaccine introduction—both among women aged 20–25 years, reflecting the childhood program, and among women aged 26–30 years, reflecting the catch-up program—showing marked reductions in CIN2+ and in lesions attributable to vaccine-covered HPV types [14,15]. An explicit evaluation of CIN3+ in women aged 25–29 years—the age group that bridges the earliest routinely vaccinated cohorts entering screening and the older catch-up cohorts—has not yet been undertaken.

Focusing on CIN3+ in the 25–29-year age group offers several advantages. CIN3+ has higher specificity than CIN2+ for clinically meaningful, treatment-relevant disease, thereby strengthening causal inference about vaccine impact. Cervical cancer screening in Norway targets women aged 25–69 years; women aged 25–29, therefore, represent those attending organized screening for the first time [16,17]. The period 2010–2024 spans the transition from an essentially unvaccinated population to one in which most women in this age group belong to vaccinated birth cohorts (≥1 dose), enabling assessment of secular trends and cohort effects across the vaccine roll-out.

Interpretation of temporal patterns must also consider changes in screening practice. In Jul 2023, Norway completed the transition from cytology every three years to primary HPV testing every five years [18]. This shift may influence detection dynamics and referral thresholds, but the direction and magnitude of any bias for CIN3+ in this narrow age group are uncertain [4]. We therefore used a histology-defined endpoint (CIN3+) and calculated rates per 1000 screened unique women—based on yearly cytology/histology files with de-duplication at the woman level—to contextualize vaccine-related declines across this programmatic transition.

This study examines trends in CIN3+ among women aged 25–29 years in Troms and Finnmark from 2010 to 2024, contrasting pre- and post-vaccination birth cohorts. The primary objective is to quantify changes in CIN3+ incidence per 1000 screened women over time and by cohort eligibility for vaccination (≥1 dose). Secondary objectives are to describe the relative contribution of routine (born ≥1997) versus catch-up (born 1991–1996) cohorts to observed trends, and to discuss the potential influence of the 2023 screening transition on measured outcomes. These findings provide a focused, policy-relevant estimate of HPV vaccine impact on the most clinically consequential precancerous endpoint at the entry to organized screening.

## 2. Materials and Methods

### 2.1. Study Design and Setting

We conducted a retrospective, population-based time-series analysis of women aged 25–29 years residing in Troms and Finnmark, Norway. The observation window spanned calendar years 2010–2024, covering the period from essentially no vaccine exposure in these age groups to the entry of routinely vaccinated cohorts into screening.

### 2.2. Data Sources

All cervical cytology and histopathology records were extracted from the laboratory information system (SymPathy; Tietoevry, Espoo, Finland) used at the Department of Clinical Pathology, University Hospital of North Norway (UNN) [14]. UNN is the sole pathology provider for Troms and Finnmark, ensuring near-complete regional capture of screening cytology and diagnostic biopsies/conizations. For descriptive context, aggregate HPV vaccination coverage (receipt of ≥1 dose) by birth cohort and county was obtained from the national immunization registry (SYSVAK), which has high completeness for the Norwegian childhood and catch-up HPV programs. Because the SymPathy dataset was anonymized for this quality-assurance study, individual-level linkage to SYSVAK was not performed; instead, vaccination exposure was assigned ecologically by birth-year eligibility and calendar year, as described below [19].

### 2.3. Study Population and Eligibility

Women were included if they had at least one cervical screening sample (cytology or HPV-based primary test) or a cervical histology during a given calendar year while aged 25–29 years. Age was calculated from the date of birth to the sample/biopsy year. Women could contribute for multiple years as they aged through the 25–29 age group. Analyses were restricted to residents of Troms and Finnmark based on the requisition information in the laboratory system [14].

### 2.4. Outcomes and Case Definitions

The primary endpoint was CIN3+, defined on histology as any of the following Norwegian pathology (NORPAT) codes [20]:CIN3: M80702Adenocarcinoma in situ (ACIS): M81402Invasive carcinoma: M80703 or M81403

When a woman had multiple histology specimens, the highest grade within a calendar year was retained. For incidence counts, each woman was counted once per year at the date of her first CIN3+ diagnosis that year.

### 2.5. Denominators and De-Duplication

For each calendar year, the denominator (“screened women”) was the number of unique women aged 25–29 with ≥1 screening sample (cytology or primary HPV test). The numerator was the number of unique women aged 25–29 with CIN3+ that year, as defined above. Multiple smears in the same year were collapsed to one record per woman; multiple biopsies were collapsed to the most severe diagnosis. If a woman had CIN3+ in a previous year, she was not re-counted as an incident case in subsequent years.

### 2.6. Cohort Classification by Vaccination Eligibility and Estimated Coverage

Cohorts were categorized by birth year to reflect vaccination opportunities: (i) routine school-based program: women born ≥1997 (offered HPV vaccination in 7. klasse at ~12 years, with high completion of the 3-dose schedule); (ii) catch-up offer: women born 1991–1996 (offered HPV vaccination as young adults during 2016–2019, with lower uptake and lower 3-dose completion than the school-based cohorts); and (iii) pre-vaccine cohorts: women born ≤1990 (no programmatic offer). Because individual vaccination status and exact number of doses were not available for all screened women, we proxied vaccination exposure at the cohort level and defined “coverage” as receipt of ≥1 HPV vaccine dose according to national reports. For the time-series, we summarized the composition of the 25–29-year age group by birth years per calendar year (e.g., 2010: 1981–1985; 2024: 1995–1999) and, assuming equal-sized birth cohorts, applied 90% coverage (≥1 dose) for birth years ≥1997 and 60% coverage (≥1 dose) for birth years 1991–1996, effective from 2017 when the first catch-up cohorts entered this age band. This yielded the following estimated proportions of vaccinated women in the 25–29-year group: 0% (2010–2016), 12% (2017), 24% (2018), 36% (2019), 48% (2020), 60% (2021), 66% (2022), 72% (2023), and 78% (2024) (Table 1). These year-specific estimates were used to interpret temporal trends in CIN3+. We could not stratify CIN3+ rates by number of doses and acknowledge this as a limitation.

### 2.7. Screening Context During the Study Period

Norway introduced primary HPV testing in a phased manner. Four counties started HPV-based screening for women aged 34–69 years in 2015, with 1:1 randomization against cytology every three years; after three years, these areas switched fully to HPV testing. At the Department of Clinical Pathology, UNN, the 50/50 randomization for women aged 34–69 years began on 1 January 2019. Additional regions were included from 2019, reaching about 50% HPV-based screening in women 34–69 years by 2022. From 1 January 2023, HPV testing was offered to women 30–69 years, and from 1 July 2023, all women 25–69 years attending screening received primary HPV testing [9]. All smears and tests were processed at UNN throughout, and histology was used to define CIN3+ to minimize modality-related variation in cytology performance. As a sensitivity description, we report annual counts of unique screened women to contextualize any changes in testing volume around the 2023 transition.

### 2.8. Statistical Analysis

For each calendar year from 2010 to 2024, we estimated:CIN3+ incidence per 1000 screened women = (number of unique women with CIN3+/number of unique women screened) × 1000.95% confidence intervals for the incidence using exact methods.

Temporal patterns were examined by linear regression with year as the independent variable. In exploratory analyses, we compared periods before and after the introduction of catch-up vaccination in the relevant age group (pre-2017 vs. 2017 onwards) and described the contribution of routinely vaccinated birth cohorts (born ≥ 1997) in 2022–2024. Statistical analyses were performed in IBM SPSS Statistics, version 29.0 (IBM Corp., Armonk, NY, USA) [21], and R, version 4.3 (R Foundation for Statistical Computing, Vienna, Austria) [22]. Two-sided *p*-values < 0.05 were considered statistically significant.

### 2.9. Ethics

The study was conducted as a quality-assurance project approved by the Regional Committee for Medical and Health Research Ethics North (REK Nord, reference 230825). It was a retrospective, registry-based assessment of existing screening and histopathology data retrieved from the SymPathy pathology system, with no intervention and no patient contact. All analyses were performed on anonymized data, and individual informed consent was not required under Norwegian regulations for quality-assurance projects.

## 3. Results

### 3.1. Screening Volume and Case Counts

From 2010 to 2024, a total of 42,253 screening tests were performed on 30,132 unique women aged 25–29 years in Troms and Finnmark. In the same period, 865 women were diagnosed with CIN3+, including 24 cases of cervical cancer (Table 2). The annual screening volume peaked in 2017 (*n* = 3410) and subsequently declined by 24.2% to 2584 tests in 2024.

### 3.2. CIN3+ Incidence over Time

The incidence of CIN3+ (per 1000 screened women) fluctuated around 15–24 in 2010–2016, rose to 26.7 per 1000 in 2017 and 27.8 per 1000 in 2018, and then declined steadily to 6.6 per 1000 in 2024 (17 cases), Table 2. This corresponds to a 75% reduction from 2017 (26.7) to 2024 (6.6). The intermediate years show a stepwise decline: 22.0 (2020), 19.0 (2021), 15.8 (2022), and 11.7 (2023).

### 3.3. Segmented Trend Analysis

To capture the pre- and post-catch-up/post-childhood vaccine era within this age group, we fitted two linear regressions to the annual CIN3+ rates: 2010–2017 and 2017–2024, Figure 1.

Thus, while the early period shows no statistically significant increase, the latter period demonstrates a clear and statistically significant annual decline in CIN3+ incidence.

### 3.4. Cervical Cancer

Cervical cancer remained infrequent throughout, with annual rates ≤ 1.5 per 1000. Peaks occurred in 2017 (5 cases; 1.5 per 1000) and 2020 (4 cases; 1.4 per 1000), while 2024 had no cases. Given the small numbers, no formal trend testing was performed for cancer.

### 3.5. Context of Screening Transition

The national shift in 2023 from cytology every three years to primary HPV testing every five years coincided with continued declines in CIN3+ rates (15.8 in 2022 to 11.7 in 2023 and 6.6 in 2024), suggesting that the downward trend persisted despite modest year-to-year variation in screening volume.

## 4. Discussion

This study documents a marked and sustained decline in CIN3+ among women aged 25–29 years in Troms and Finnmark from 2017 onward, following an initial period without a significant trend during 2010–2017. Segmented regression showed no statistically significant change before 2017, but a clear annual reduction thereafter (−3.04 per 1000 screened women per year), coinciding with the gradual replacement of unvaccinated women by birth years offered HPV vaccination. The annual CIN3+ rate fell from 26 to 28 per 1000 in 2017–2018 to 6.6 per 1000 in 2024, representing a ~75% relative decline. Cervical cancer remained rare throughout, with small numbers precluding formal trend testing. In the Norwegian Cervical Cancer Screening Programme, the number of women aged 25–27 treated with conization declined from 1360 in 2017 to 703 in 2024 (−48.3%), a sustained decrease that parallels the sharp fall in CIN3+ in our material and indicates a substantial reduction in excisional treatment at reproductive age, with likely benefits for obstetric outcomes and healthcare utilization [23].

These findings are in line with, and extend, previously reported vaccine effects on high-grade cervical disease from the Nordic region. In the 14-year Nordic long-term follow-up of the FUTURE II trial, no HPV16/18-related CIN2+ cases were detected, demonstrating durable 100% protection against vaccine-type high-grade precancer [7]. At the population level, Danish data have shown real-world reductions of about 49% in CIN2+ and 47% in CIN3+ after introduction of the quadrivalent vaccine, i.e., for lesions irrespective of HPV type [8]. Our observed ~75% decline in CIN3+ among 25–29-year-old screened women in Northern Norway is therefore consistent with these reports, especially considering that, by 2024, most women in this age band had been vaccinated before sexual debut and that we used a histology-defined endpoint. Moreover, nationwide studies from Denmark and Sweden have demonstrated that such reductions in high-grade precancer translate into lower invasive cervical cancer incidence, with approximately 86% (Denmark) and 88% (Sweden) lower risk among women vaccinated before 17 years of age [9,10]. The temporal pattern in our material—minimal change while unvaccinated and partly vaccinated birth years dominated, followed by a steep decline when fully vaccinated birth years entered screening—is compatible with this biological and epidemiological gradient by age at vaccination.

Our earlier regional analyses showed vaccine-associated declines in high-grade disease among women aged 20–25 years (reflecting the school-based program) and 26–30 years (reflecting the catch-up program) [14,15]. The present analysis adds two elements: first, it isolates CIN3+ as the outcome, which has higher specificity for clinically relevant precancer than CIN2+; second, it focuses on the 25–29-year age group, where women vaccinated in the school-based program and women vaccinated later in the catch-up campaign now meet in screening. The timing and magnitude of the decline mirror the changing vaccination mix in this age group, with the proportion vaccinated rising from about 12% in 2017 to nearly 80% by 2024. As reported from Denmark and Sweden, the largest effects are seen in women vaccinated at a young age, whereas those vaccinated later in Norway show a more modest impact, consistent with lower uptake and more HPV exposure before vaccination. This gradient by age at vaccination parallels what has been observed for cervical cancer and supports a causal interpretation of the declines.

An actual reduction in invasive cervical cancer in these birth years cannot yet be quantified from our data because women born 1995–1999 were only 25–29 years old in 2020–2024, and cervical cancer at these ages is rare even without vaccination; our series contained too few cases to model trends. This lag is expected: vaccination occurred in adolescence, high-grade precancer appears several years later, and cancer typically arises even later. The Danish and Swedish nationwide studies showed measurable reductions in cervical cancer only after a decade or more of follow-up, and mainly among women vaccinated before age 17, with risk reductions of ≈86% and ≈88%, respectively, compared with unvaccinated women [9,10]. It is therefore reasonable to infer that the 75% decline in CIN3+ we observed at entry to screening should translate into a substantial fall in cervical cancer in the same vaccinated birth years once they reach their 30s and 40s, provided screening attendance remains high.

To strengthen and refine these observations, three complementary lines of work are needed: (i) individual-level linkage of vaccination, screening, HPV typing, and histology to confirm that the decline is concentrated in vaccinated women and mainly involves vaccine-targeted types; (ii) longer follow-up of the current 25–29-year-olds to determine when a statistically robust reduction in invasive cervical cancer becomes detectable; and (iii) surveillance for possible shifts toward non-vaccine HPV types in CIN3+ as HPV-based screening is fully implemented. Such studies would place these regional Norwegian data in line with Danish and Swedish cancer-endpoint evidence and allow more precise estimates of vaccine-preventable disease in high-coverage settings.

Interpretation of these trends must also consider contemporaneous changes in screening. Norway completed the national transition in July 2023 from cytology every three years to primary HPV testing every five years [9]. In theory, this should increase or at least maintain detection because HPV testing is more sensitive than cytology and can alter referral dynamics. In our material, however, CIN3+ rates continued to decline from 2022 to 2023 and further to 2024—i.e., after HPV testing was introduced for 25–29-year-olds—despite the change in primary test. Because our endpoint was histology-defined (CIN3+) and rates were expressed per 1000 unique screened women, the estimates are relatively robust to shifts in cytology performance or test sensitivity. Taken together, the temporal pattern—an early plateau, a peak around 2017–2018, followed by a sustained decline—aligns more closely with the increasing proportion of vaccinated birth cohorts entering screening than with screening artifacts, indicating that the fall in CIN3+ is mainly vaccine-driven across the screening transition.

We also note a 24% reduction in the number of screening tests from 2017 to 2024. This should not be interpreted as a deterioration of the programme but as a contextual factor. Part of the higher screening activity earlier in the period likely reflected temporary, media-driven demand for extra cytology in young women, while from 2023 the programme shifted to primary HPV testing, which may have normalized attendance [24]. Because our outcome is histology-defined CIN3+ expressed per 1000 screened unique women, the observed decline is unlikely to be explained by lower screening volume alone and is more plausibly driven by the increasing proportion of vaccinated cohorts entering screening.

Taken together, our data are concordant with clinical trial evidence (durable protection against HPV16/18-related CIN2+), with Nordic real-world data on reductions in CIN2+/CIN3+, and with national registry studies from Denmark and Sweden showing that early HPV vaccination also prevents invasive cervical cancer. In this context, the ~75% reduction in CIN3+ we observed among 25–29-year-old screened women in Northern Norway represents the expected effect size when highly vaccinated birth years start to dominate the screening-eligible population.

Mechanistically, this pattern fits with the gradual entry into screening of women who received HPV16/18-containing vaccines and with the genotype profile expected after the school-based program. In Northern Norway, high-grade lesions in vaccinated women were rarely HPV16/18-positive, consistent with strong direct protection and little short- to medium-term type replacement [5,6]. The weaker effect in women vaccinated later through the catch-up campaign is consistent with lower uptake and more HPV exposure before vaccination. Together, these observations support using histology-defined CIN3+ per 1000 screened women as a conservative indicator of vaccine impact and highlight the need for ongoing surveillance of non-vaccine HPV types.

This analysis has several strengths: it is population-based within a clearly defined catchment; all cervical samples were processed at a single pathology department; and standardized NORPAT codes with de-duplication at the woman level improved case ascertainment. The long observation window (2010–2024) spans the transition from mostly unvaccinated to predominantly vaccinated screening participants, allowing robust time-series assessment. Our endpoint—CIN3+—is clinically meaningful and less prone to misclassification than CIN2 [25]; in previous regional series from Northern Norway, the relative declines were larger for CIN3+ than for CIN2+, underscoring the suitability of CIN3+ for evaluating HPV vaccine impact [15,16].

Beyond diagnostic endpoints, procedural burden declined in parallel: conization (LEEP/LLETZ) rates among young women fell substantially over time, consistent with fewer CIN3+ diagnoses and indicating downstream benefits for quality of care, fertility preservation, and resource use [15,16,23].

This is feasible in Norway because linkage-ready infrastructure is already in place: unique personal identifiers, the SymPathy LIS capturing regional cytology and histology, and established connections to SYSVAK and the Cancer Registry. These sources can be combined to identify rare breakthrough lesions in vaccinated women, to characterize the HPV type distribution in post-vaccination CIN3+, and to strengthen causal attribution in this age group [19,20].

Limitations merit consideration. Vaccination coverage was available at the cohort level (from SYSVAK) and was mapped onto the annual birth-year composition of the 25–29-year group, which is appropriate for an anonymized, register-based quality study but does not allow individual-level vaccine effectiveness estimates or assessment of within-cohort heterogeneity. Furthermore, HPV genotype data were not available for the CIN3+ biopsies because HPV testing was not part of primary screening in this age group before 2023, and biopsies are not routinely genotyped. We therefore infer, rather than demonstrate, that the decline is largely driven by vaccine-targeted types, consistent with national Norwegian monitoring. Finally, with increasing coverage, part of the observed reduction may reflect herd effects among unvaccinated women.

Second, rates were expressed per 1000 screened women rather than as population incidence; variation in screening attendance could therefore affect denominators, although the decline persisted despite fluctuating annual volumes. Third, CIN3+ was not stratified by HPV genotype, so the contribution of vaccine-targeted types could not be demonstrated directly; future linkage with HPV typing will clarify type-specific impacts. Fourth, program changes (referral thresholds, colposcopy practice) and broader secular trends may act as unmeasured confounders, but the size and timing of the decline are most parsimoniously explained by vaccination. Finally, invasive cervical cancer remained rare in this age group, limiting formal trend analysis, but the near-absence of cancer in vaccinated cohorts is directionally consistent with the observed fall in CIN3+ and with the expected lag from precancer to invasive disease.

Taken together, the data indicate that the roll-out and increasing coverage of HPV vaccination in Norway—especially routine vaccination before sexual debut—has translated into substantial real-world reductions in CIN3+ among women entering organized screening. As vaccinated birth years continue to replace earlier, largely unvaccinated birth years, and provided screening attendance remains stable, further declines are likely. Future work should link individual vaccination records with screening and histology, assess HPV type–specific CIN3+, and follow the post-2023 HPV-based screening era to determine when a statistically robust reduction in invasive cervical cancer becomes observable in these vaccinated birth years.

## 5. Conclusions

Among women aged 25–29 years in Troms and Finnmark, CIN3+ incidence was relatively stable up to 2017 and then declined consistently through 2024. This temporal pattern mirrors the increasing share of vaccinated birth cohorts within the age group and appears unaffected by the 2023 transition to primary HPV testing, supporting a genuine vaccine impact on clinically significant precancer at the age of entry to organized screening. Sustaining high HPV vaccination uptake—especially before sexual debut—together with equitable catch-up and continued surveillance will be essential to maintain these gains.

## Figures and Tables

**Figure 1 vaccines-13-01147-f001:**
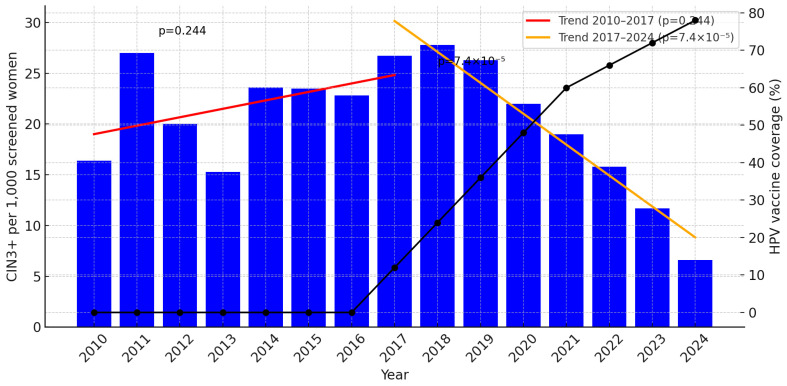
Incidence of CIN3+ per 1000 screened women aged 25–29 years in Troms and Finnmark, 2010–2024, with estimated HPV vaccination coverage (%) plotted on the right Y-axis. Segmented linear regression (red: 2010–2017; orange: 2017–2024) shows a non-significant increase in 2010–2017 (slope +0.83 per 1000 per year; *p* = 0.244) and a significant decline in 2017–2024 (slope −3.04 per 1000 per year; *p* = 7.4 × 10^−5^). The progressive increase in vaccination coverage from 2017 to 2024 (black line) parallels the decline in CIN3+. For year-specific estimates of vaccinated women, see Table 1.

**Table 1 vaccines-13-01147-t001:** Estimated HPV vaccine coverage among women aged 25–29 by calendar year (Troms and Finnmark), 2010–2024.

Year	Birth Cohorts (25–29)	Estimated Coverage (%)
2010	1981–1985	0
2011	1982–1986	0
2012	1983–1987	0
2013	1984–1988	0
2014	1985–1989	0
2015	1986–1990	0
2016	1987–1991	0
2017	1988–1992	12
2018	1989–1993	24
2019	1990–1994	36
2020	1991–1995	48
2021	1992–1996	60
2022	1993–1997	66
2023	1994–1998	72
2024	1995–1999	78

**Table 2 vaccines-13-01147-t002:** Number of screening tests and incidence of CIN3+ and cervical cancer (per 1000 screened women) among women aged 25–29 years in Troms and Finnmark, Norway, 2010–2024.

Year	Screening Tests	CIN3+	CIN3+ Rate (95% CI)	Cancer	Cancer Rate
2010	2257	37	16.4 (11.6–22.5)	2	0.9
2011	2294	62	27.0 (20.8–34.5)	2	0.9
2012	2246	45	20.0 (14.7–26.7)	0	0.0
2013	2486	38	15.3 (10.8–20.9)	1	0.4
2014	2627	62	23.6 (18.1–30.2)	1	0.4
2015	2811	66	23.5 (18.2–29.8)	1	0.4
2016	3021	69	22.8 (17.8–28.8)	1	0.3
2017	3410	91	26.7 (21.5–32.7)	5	1.5
2018	3168	88	27.8 (22.3–34.1)	1	0.3
2019	2963	78	26.3 (20.9–32.7)	3	1.0
2020	2909	64	22.0 (17.0–28.0)	4	1.4
2021	3318	63	19.0 (14.6–24.2)	1	0.3
2022	3166	50	15.8 (11.7–20.8)	1	0.3
2023	2993	35	11.7 (8.2–16.2)	1	0.3
2024	2584	17	6.6 (3.8–10.5)	0	0.0
Total	42,253	865	20.5 (19.1–21.9)	24	0.6

## Data Availability

The raw data supporting the conclusions of this article are available from the corresponding author on reasonable request, subject to privacy and institutional restrictions.

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
