# Peer review of "HPV Vaccination and CIN3+ Among Women Aged 25–29 Years in Northern Norway, 2010–2024: A Population-Based Time-Series Analysis"

_vaccines, 2025, doi:10.3390/vaccines13111147_

Round 1

Reviewer 1 Report

Comments and Suggestions for Authors

This is a well-written paper summarizing important data about the protection afforded by HPV vaccine.  As the authors state, this study looks at the "most clinically consequential precancerous endpoint" for HPV.  

GENERAL COMMENTS

The Introduction needs to describe in a paragraph or two what is known about vaccine efficacy/effectiveness from other research to date, including protection against CIN2+, CIN3+, and cervical cancer.   Similarly, the Discussion should compare the results of this study to those data. 

It is unclear exactly what estimated vaccine coverage means in this study & what the country specific vaccination rates reflect (such as, but not limited to lines 41 & 129.)  Is it one or more doses or the full 2 or 3-dose series depending on age.  Please clarify in the Introduction, Abstract, Methods, and Results.  Importantly, please describe whether the results varied by the number of doses received.  If this cannot be done for some reason, list as a limitation of the study. 

Clarify which HPV vaccine was used in the study (2, 4, or 9 valent) & the standard regimen for the study population by age(s).   This is particularly important to put some of the Discussion (lines 227/234) in context.  

SPECIFIC COMMENTS

Line 27 - please use a different word other than "maturation" of HPV vaccination here & elsewhere in the manuscript.  Is this the development of the HPV program across various ages, the vaccine uptake, or something else. (Usually this term is used in vaccines to refer to the maturation of the immune response.)  

Line 29 - specify the age of entry to organized screening in the Abstract.   

Line 41 - define vaccination coverage rate based on number of doses received.  and consider adding another time point to this list - perhaps vaccination coverage to middle cohorts (with the years specified for each time point).

Lines 55-63 - nice explanation - very important.

Table 1.  Consider merging with Figure 1 so the reader can easily see the trends in vaccination rates vs the trends in CIN3+.  Vaccination rate might be another line across the top (with the axis on the right Y axis). 

Line 137 - specify when the transition started and ended (20XX-2023).

Line 155 - add a very brief explanation of a quality project.  Does this mean data analysis with no intervention, no consent needed?

Lines 163-164 - any explanation for the 24% drop in screening?  If so, please add. 

Line 168-169 - why present decrease from 2018-2024 & 2017-2024?  Suggest just present one value. 

Figure 1 -  consider merging with Table 1 as described earlier.

Discussion - discuss when estimates of actual reduction in cancer might be expected from this data set & how these results compare to other studies that have evaluated CIN3+ or cancer as an endpoint.  Describe any other studies that should be conducted to support or extend these results.   This is an important part of any Discussion. .  

Lines 289-292 & 299-302 - text is repetitive.  Suggest just putting text regarding sustained high vaccination uptake in the Conclusions & leave sentence about future work in the Discussion (but expand).

Line 221 - define partial uptake (using a %).

Line 227-228 - use other words other than "cohort maturation" and "vaccine type specificity".  What do these terms really mean & what point is being made?

Lines 228-233 - assume a 16/18 HPV vaccine was used, but not clear else where.  This text is a bit hard to follow.  Consider rewriting.  What is a" policy relevant endpoint" (lines 232 & 248)?  Perhaps just relevant endpoint?

Lines 261-262 - why was vaccination coverage estimated at the cohort level & not at the individual level?  How good is it?  This may warrant further explanation in the Methods or here.

Author Response

Reviewer 1

Comments 1: This is a well-written paper summarizing important data about the protection afforded by HPV vaccine.  As the authors state, this study looks at the "most clinically consequential precancerous endpoint" for HPV.  

Response 1: We thank the reviewer for the positive assessment. We agree that CIN3+ represents the most clinically consequential precancerous endpoint for evaluating the impact of HPV vaccination.

Comments 2: The Introduction needs to describe in a paragraph or two what is known about vaccine efficacy/effectiveness from other research to date, including protection against CIN2+, CIN3+, and cervical cancer.

Response 2: We have expanded the Introduction to summarize current evidence on HPV vaccine effectiveness. We now cite the Nordic long-term follow-up of the FUTURE II trial by Kjaer et al. demonstrating sustained 100% effectiveness against HPV16/18-related CIN2+ for up to 12–14 years after vaccination (EClinicalMedicine 2020). We have also added Danish population-based data showing real-world reductions of 49% in CIN2+ and 47% in CIN3+ after introduction of the quadrivalent vaccine, i.e. for lesions irrespective of HPV type. In addition, nationwide studies from Denmark (Kjaer et al., JNCI 2021) and Sweden (Lei et al., N Engl J Med 2020) reported ≈86% and ≈88% lower incidence of invasive cervical cancer, respectively, among women vaccinated before age 17. Together, these data show that prophylactic HPV vaccination prevents the high-grade precursors (CIN2+/CIN3+) and that this protection translates into measurable reductions in cervical cancer, which supports our focus on CIN3+ in the present time-series analysis.

Comments 3: Similarly, the Discussion should compare the results of this study to those data. 

Response 3: We have updated the Discussion to compare our findings with Nordic trial-based long-term follow-up data, Danish population-based reductions in CIN2+/CIN3+, and nationwide Danish and Swedish studies demonstrating large decreases in invasive cervical cancer among women vaccinated before age 17.

Comments 4: It is unclear exactly what estimated vaccine coverage means in this study & what the country specific vaccination rates reflect (such as, but not limited to lines 41 & 129.)  Is it one or more doses or the full 2 or 3-dose series depending on age.  Please clarify in the Introduction, Abstract, Methods, and Results.  Importantly, please describe whether the results varied by the number of doses received.  If this cannot be done for some reason, list as a limitation of the study.

Response 4: In this study, “estimated vaccine coverage” refers to the proportion of women in the relevant birth cohorts who had received at least one dose of HPV vaccine according to national immunization registry data. For the school-based program (girls in 7th grade, born 1997 or later), uptake of the full 3-dose series was high, so ≥1 dose closely approximates full vaccination in these cohorts. In the catch-up program (women born 1991–1996, vaccinated 2016–2019), uptake of ≥1 dose was lower and completion of all 3 doses was somewhat lower than in the school-based cohorts. From 2018, a 2-dose schedule was introduced for girls <15 years, but these cohorts have not yet entered the 25–29-year screening age and are therefore not part of the present analysis. We did not have individual-level dose information for all screened women and could not analyse CIN3+ by number of doses; this is now stated as a limitation.

Comments 5: Clarify which HPV vaccine was used in the study (2, 4, or 9 valent) & the standard regimen for the study population by age(s).   This is particularly important to put some of the Discussion (lines 227/234) in context.  

Response 5: Norway introduced HPV vaccination for 12-year-old girls in 2009 (birth cohorts ≥1997) with the quadrivalent vaccine (Gardasil, 3-dose schedule). In the national catch-up program in 2016–2019 for women born 1991–1996, the bivalent vaccine (Cervarix, 3-dose schedule) was used. In 2018, the childhood program changed from quadrivalent to bivalent vaccine, but those younger cohorts have not yet reached the 25–29-year screening age and are therefore not included in the present analysis.

Comments 6: Line 27 - please use a different word other than "maturation" of HPV vaccination here & elsewhere in the manuscript.  Is this the development of the HPV program across various ages, the vaccine uptake, or something else. (Usually this term is used in vaccines to refer to the maturation of the immune response.)  

Response 6: We have reformulated all sentences that used the term “maturation” of HPV vaccination to clarify that we refer to the roll-out of the program and the progressive entry of vaccinated birth cohorts into the screening age, in order to avoid confusion with immune response maturation.

Comments 7: Line 29 - specify the age of entry to organized screening in the Abstract.   

Response 7: In Norway, cervical cancer screening is recommended from 25 to 69 years of age, and we have updated the Abstract to specify that the observed decline in CIN3+ occurred at the age of entry to organized screening (25–29 years).

Comments 8: Line 41 - define vaccination coverage rate based on number of doses received.  and consider adding another time point to this list - perhaps vaccination coverage to middle cohorts (with the years specified for each time point).

Response 8: Vaccination coverage in this manuscript is defined as receipt of ≥1 dose of HPV vaccine. In the school-based program (girls in 7th grade, birth cohorts 1997 and later), completion of the 3-dose schedule was high, so ≥1 dose closely reflects full vaccination. We have clarified this in the Introduction and expanded the description as follows: “Norway introduced HPV vaccination in 2009 for girls in the 7th grade (birth cohorts 1997 and later), with coverage (≥1 dose) increasing from about 70% in the earliest cohorts (2009–2011) to around 85% in the mid-cohorts and to over 90% in recent years. In addition, girls born in 1997 who did not receive HPV vaccination in 2009 were offered vaccination in the 2016–2019 catch-up program, which further increased coverage in this birth cohort.”

Comments 9: Lines 55-63 - nice explanation - very important.

Response 9: Thank you for this positive comment.

Comments 10: Table 1.  Consider merging with Figure 1 so the reader can easily see the trends in vaccination rates vs the trends in CIN3+.  Vaccination rate might be another line across the top (with the axis on the right Y axis). 

Response 10: Thank you for the suggestion. We have now added the estimated HPV vaccination coverage as a separate line (right Y-axis) in the figure showing annual CIN3+ per 1,000 screened women (2010–2024), so the two trends can be viewed together. We have kept Table 1 in the manuscript for reference.

Comments 11: Line 137 - specify when the transition started and ended (20XX-2023).

Response 11: Norway transitioned from cytology-based screening every three years to primary HPV testing every five years in a stepwise manner: HPV testing was first introduced in 2015 in four counties for women aged 34–69 years with 1:1 randomization against cytology; after three years these counties switched fully to HPV testing. Additional regions started the same phased model from 2019, reaching about 50% HPV-based screening in women 34–69 years by 2022. From 1 Jan 2023 HPV testing was extended to women 30–69 years, and from 1 Jul 2023 all women aged 25–69 years attending screening received primary HPV testing.

Comments 12: Line 155 - add a very brief explanation of a quality project.  Does this mean data analysis with no intervention, no consent needed?

Response 12: Yes. The analysis was conducted as a quality-assurance project approved by the Regional Committee for Medical and Health Research Ethics North (REK Nord, reference 230825). This was a retrospective, register-based evaluation of existing screening and histology data from the SymPathy pathology system, with no intervention and no contact with patients. All analyses used anonymized records; informed consent was therefore not required under Norwegian regulations for quality-assurance studies.

Comments 13: Lines 163-164 - any explanation for the 24% drop in screening?  If so, please add. 

Response 13: The temporary increase in screening activity in 2016-2018 and 2021–2022 was likely driven by extensive national media coverage of cervical cancer cases in young women after false-negative Pap smears, which prompted many women to request additional cytology outside the routine schedule. In 2023, primary HPV testing was introduced also for women aged 25–33 years, media attention declined, and no cervical cancer cases were reported among HPV-vaccinated 25-year-old women in 2022, which may have reduced anxiety and the perceived need for extra tests. Consequently, screening numbers in 2024 were lower, reflecting a return toward routine screening rather than a true decline in program performance.

Comments 14: Line 168-169 - why present decrease from 2018-2024 & 2017-2024?  Suggest just present one value. 

Response 14: We agree. We have kept the decline from 2017 to 2024 (26.7 to 6.6 per 1,000; ≈75% reduction) and removed the 2018–2024 figure from the Results.

Comments 15: Figure 1 -  consider merging with Table 1 as described earlier.

Response 15: Based on this comment, we have updated Figure 1 to include the estimated HPV vaccination coverage (right Y-axis) using the values from Table 1, so that trends in coverage and CIN3+ can be viewed together. We have retained Table 1 as a reference for the exact year-specific coverage estimates.

Comments 16: Discussion - discuss when estimates of actual reduction in cancer might be expected from this data set & how these results compare to other studies that have evaluated CIN3+ or cancer as an endpoint.  Describe any other studies that should be conducted to support or extend these results.   This is an important part of any Discussion.  

Response 16: We have updated the Discussion to explain that invasive cervical cancer is still too rare in this 25–29-year-old, recently vaccinated population to estimate a trend, to relate our CIN3+ findings to Danish and Swedish nationwide data showing 86–88% cancer reduction after early vaccination, and to outline future studies (individual-level linkage, longer follow-up, and type-specific monitoring) needed to confirm and extend these results.

Comments 17: Lines 289-292 & 299-302 - text is repetitive.  Suggest just putting text regarding sustained high vaccination uptake in the Conclusions & leave sentence about future work in the Discussion (but expand).

Response 17: Thank you for noting the repetition. This occurred when transferring the manuscript to the MDPI Vaccines template. We have now removed the duplicate passage, kept the statement on sustained high vaccination uptake in the Conclusions, and retained/expanded the sentence on future work (individual-level linkage, type-specific CIN3+, and longer-term cancer outcomes) in the Discussion.

Comments 18: Line 221 - define partial uptake (using a %).

Response 18: We have clarified this sentence to read: “In contrast, the catch-up cohorts (born 1991–1996) likely experienced attenuated effectiveness due to partial uptake (≈60% received ≥1 dose, compared with ≈90% in the school-based cohorts) and prior HPV exposure before vaccination.”

Comments 19: Line 227-228 - use other words other than "cohort maturation" and "vaccine type specificity".  What do these terms really mean & what point is being made?

Response 19: We have reformulated this part of the Discussion to state explicitly that the decline follows the stepwise entry of vaccinated birth cohorts into the 25–29-year screening group and that the observed drop in HPV16/18-positive lesions is consistent with the vaccines targeting these types, rather than using the less precise terms “cohort maturation” and “vaccine-type specificity.”

Comments 20: Lines 228-233 - assume a 16/18 HPV vaccine was used, but not clear else where.  This text is a bit hard to follow.  Consider rewriting.  What is a" policy relevant endpoint" (lines 232 & 248)?  Perhaps just relevant endpoint?

Response 20: We have clarified that the school-based cohorts received the quadrivalent HPV 6/11/16/18 vaccine and that the catch-up cohorts received the bivalent HPV 16/18 vaccine. We also rewrote the paragraph to avoid the term “policy-relevant endpoint” and now describe CIN3+ simply as a clinically relevant, less misclassified endpoint for evaluating HPV vaccine impact.

Comments 21: Lines 261-262 - why was vaccination coverage estimated at the cohort level & not at the individual level?  How good is it?  This may warrant further explanation in the Methods or here.

Response 21: We have clarified in the Methods that vaccination exposure was derived from SYSVAK birth year– and county–specific coverage (≥1 dose) and applied to the 25–29-year age band for each calendar year, because the screening/histology dataset was anonymized and could not be individually linked to vaccination records. In the Limitations, we now state explicitly that this cohort-level approach is based on high-quality national data and is suitable for ecological trend analysis, but it does not allow estimation of individual-level vaccine effectiveness and may conceal within-cohort heterogeneity.

Reviewer 2 Report

Comments and Suggestions for Authors

The paper by Sorbye et al. presents a cohort study evaluating the impact of HPV vaccination on the incidence of CIN3+ in women aged 25-29 years between 2010 and 2024 in Northern Norway. This chosen period of 15 years is interesting because it includes a pre-vaccination period from 2010 to 2017 and a period of immunization with progressive increase of vaccine coverage from 2019 to 2024. The methodology of the study is robust, the presentation of results is clear and the conclusions are in perfect agreement with the presented data.

Given the retrospective and anonymous design of the study, it was not possible to correlate more closely the occurrence of CIN3+ lesions with the vaccinal status and the time at which the vaccine was administered if any. In addition, no data are available concerning the genotypes of HPV involved in the positive cases; if available, this information is probably easy to collect and could be useful to be shown for discussing less theoretically the influence of this parameter in the post-vaccination era by comparison to the pre-vaccination period.

In addition to the previous comment, I have a few more remarks to formulate for improving the quality of the manuscript that is already excellent:

  • for people not trained with Norwegian practices in terms of HPV immunization, it would be useful to recall which kind(s) of vaccines is/are used in this country, including the genotypes covered by the vaccination;
  • in the Results section, it would be useful to know the total number of women who were involved in this study since a part of them were tested several times;
  • Figure 1 is the central message of the paper; however, it is not useful to report three times the same information relative to the results: above the figure, on the figure and in the caption. The colored text on the Figure is difficult to read and can be omitted;
  • the Discussion section is too centered on the results presented above concerning this North part of Norway. It could be useful to know whether similar data of vaccine efficiency are available for other Norwegian regions, Scandinavian countries and other countries across the world. It would help to better illustrate the added value of this very interesting study.

Author Response

Reviewer 2

Comments 1: The paper by Sorbye et al. presents a cohort study evaluating the impact of HPV vaccination on the incidence of CIN3+ in women aged 25-29 years between 2010 and 2024 in Northern Norway. This chosen period of 15 years is interesting because it includes a pre-vaccination period from 2010 to 2017 and a period of immunization with progressive increase of vaccine coverage from 2019 to 2024. The methodology of the study is robust, the presentation of results is clear and the conclusions are in perfect agreement with the presented data.

Response 1: Thank you for the positive evaluation of our study design, results, and conclusions.

Comments 2: Given the retrospective and anonymous design of the study, it was not possible to correlate more closely the occurrence of CIN3+ lesions with the vaccinal status and the time at which the vaccine was administered if any. In addition, no data are available concerning the genotypes of HPV involved in the positive cases; if available, this information is probably easy to collect and could be useful to be shown for discussing less theoretically the influence of this parameter in the post-vaccination era by comparison to the pre-vaccination period.

Response 2: We agree that individual linkage to HPV vaccination status and HPV genotype would strengthen the analysis. In this retrospective, anonymized dataset from the SymPathy pathology system we could not link women to SYSVAK or retrieve HPV genotypes for each CIN3+ case. In addition, HPV testing was not part of routine primary screening for women aged 25–29 years before 2023, and cervical biopsies with CIN3 are not routinely genotyped in clinical practice. We therefore discussed genotype patterns on the basis of national monitoring data. In Norway, the Norwegian Cancer Registry and the Norwegian Institute of Public Health run a national quality-assurance project that links screening data with SYSVAK and retrospectively HPV-types paraffin-embedded tissue from cervical cancer, ACIS and a sample of CIN2/CIN3 cases; these data already show marked reductions in HPV16/18 after vaccination. We have now clarified this in the Discussion and noted the lack of individual linkage and genotype data as a limitation.

Comments 3: for people not trained with Norwegian practices in terms of HPV immunization, it would be useful to recall which kind(s) of vaccines is/are used in this country, including the genotypes covered by the vaccination;

Response 3: In Norway, HPV vaccination was introduced in 2009 for 12-year-old girls (birth cohorts 1997 and later) with the quadrivalent vaccine (Gardasil), which protects against HPV types 6, 11, 16, and 18. A national catch-up programme in 2016–2019 offered vaccination to women born 1991–1996 with the bivalent vaccine (Cervarix), which protects against HPV types 16 and 18. Gardasil 9 is available privately but is not part of the publicly funded programme and has limited uptake. Thus, women aged 25–29 years in our study were predominantly vaccinated with either quadrivalent (6/11/16/18) or bivalent (16/18) vaccine, and the observed decline in CIN3+ mainly reflects prevention of lesions caused by HPV 16/18.

Comments 4: in the Results section, it would be useful to know the total number of women who were involved in this study since a part of them were tested several times

Response 4: Thank you for your comment. We agree that clarifying this point is useful. In Norway, women were recommended to undergo cervical screening every three years, starting at age 25. As such, women who began screening at age 25 were due for their next screening at age 28. In total, 42,253 screening tests were conducted, involving 30,132 unique women, some of whom were tested multiple times during the study period.

Comments 5: Figure 1 is the central message of the paper; however, it is not useful to report three times the same information relative to the results: above the figure, on the figure and in the caption. The colored text on the Figure is difficult to read and can be omitted

Response 5: Response 4: We agree. Figure 1 has been updated: we removed the text above the figure and revised the caption to describe the CIN3+ trends, the two regression segments, and the HPV vaccination coverage line.

Comments 6: the Discussion section is too centered on the results presented above concerning this North part of Norway. It could be useful to know whether similar data of vaccine efficiency are available for other Norwegian regions, Scandinavian countries and other countries across the world. It would help to better illustrate the added value of this very interesting study.

Response 6: We have expanded the Discussion to situate our regional findings within the broader Nordic and national context. We now refer to Norwegian registry monitoring showing marked reductions in HPV16/18 and high-grade lesions in vaccinated cohorts, and we cite population-based data from Denmark and Sweden demonstrating large decreases in CIN2+/CIN3+ and, for women vaccinated before age 17, 86–88% lower cervical cancer incidence. Since Norway has used the same vaccines and similar school-based delivery, these external data support and extend the effect size observed in Northern Norway.

Reviewer 3 Report

Comments and Suggestions for Authors

By analyzing the HPV vaccination status and CIN3+ cases of 25-29-year-old women in northern Norway from 2010 to 2024, it was concluded that maintaining a high HPV vaccination rate before the first sexual activity, fair catch-up vaccination, and continuous monitoring are the keys to maintaining a steady decline in the clinically significant precancerous lesion indicator - CIN3+ and keeping it at a low level. This indicates that the HPV vaccination strategy, monitoring methods, monitoring intervals, and the case determination standard - CIN3+ are all scientific and correct, and should be continued.

To improve the article, the following points need to be addressed:

  1. In the introduction, the association between HPV and cervical cancer, the advantages and disadvantages of cytology and HPV testing (including sensitivity, specificity, and correlation with cases) should be introduced.
  2. If only the monitoring period changes from 3 years to 5 years, the detection rate based on the number of monitored cases should not change significantly. However, if the monitoring method changes from cytology to HPV testing, the detection rate may change. Therefore, a comparison test - the coincidence rate between cytology and HPV testing or the correlation of each with CIN3+ cases - should be added.
  3. The annual incidence of cervical cancer is in single digits. It can be traced whether the cervical cancer patients have not been vaccinated or have not received catch-up vaccination, which can better prove the preventive effect of vaccination on CIN3+ cases and cervical cancer patients.
  4. This study only investigated the HPV vaccination status and CIN3+ cases of 25-29-year-old women. The incidence of CIN3+ cases or cervical cancer and its relationship with non-vaccination among women aged 29 and above until death should also be investigated.
  5. Since 2010, the 12-year-old girls who received HPV vaccination in 1997 have entered the lower limit of the monitoring age of 25. If CIN3+ cases or cervical cancer patients occur among the vaccinated girls monitored from this time, it may be due to HPV type mismatch, which also verifies the efficacy of the HPV vaccine from another aspect.
  6. The "1,2" after the author Elin Synnøve Mortensen should be superscript.

Author Response

Reviewer 3

Comments 1: By analyzing the HPV vaccination status and CIN3+ cases of 25-29-year-old women in northern Norway from 2010 to 2024, it was concluded that maintaining a high HPV vaccination rate before the first sexual activity, fair catch-up vaccination, and continuous monitoring are the keys to maintaining a steady decline in the clinically significant precancerous lesion indicator - CIN3+ and keeping it at a low level. This indicates that the HPV vaccination strategy, monitoring methods, monitoring intervals, and the case determination standard - CIN3+ are all scientific and correct, and should be continued.

Response 1: Thank you for the positive assessment and for highlighting the importance of high coverage before sexual debut, equitable catch-up, and continuous monitoring.

Comments 2: In the introduction, the association between HPV and cervical cancer, the advantages and disadvantages of cytology and HPV testing (including sensitivity, specificity, and correlation with cases) should be introduced.

Response 2: Cervical cancer is caused by persistent infection with oncogenic HPV types, most commonly HPV 16 and 18. Cervical carcinogenesis usually develops over many years through precursor lesions (CIN2/3), which makes screening an effective prevention strategy. Conventional cytology-based screening aims to detect and treat these precancers before invasion, but cytology has only moderate sensitivity (about 50–70%), so cervical cancer can still occur in women with repeatedly normal smears. HPV-based screening is more sensitive than cytology and has been shown to provide about 60–70% better protection against invasive cervical cancer than cytology-based screening, but at the cost of lower specificity, more HPV-positive tests, and a lower positive predictive value for CIN3+. Because of this, HPV-positive women require triage to avoid unnecessary colposcopy and biopsies; cytology is still the most used triage method, but partial genotyping (HPV16/18), extended genotyping, and dual staining (p16/Ki-67) are also used in many programs.

Comments 3: If only the monitoring period changes from 3 years to 5 years, the detection rate based on the number of monitored cases should not change significantly. However, if the monitoring method changes from cytology to HPV testing, the detection rate may change. Therefore, a comparison test - the coincidence rate between cytology and HPV testing or the correlation of each with CIN3+ cases - should be added.

Response 3: In Norway, women aged 25–29 years were not included in primary HPV-based screening before 2023, so we could not perform a direct cytology–HPV comparison in this age group within our dataset. Because HPV testing is more sensitive than cytology, a switch to HPV as the primary test would normally be expected to maintain or increase CIN3+ detection. In our material, however, CIN3+ rates continued to fall from 2022 to 2023 and further to 2024 despite the 2023 switch to HPV testing. This supports that the decline is driven by increasing HPV vaccination coverage in the screened cohorts, rather than by a change in screening method. We have clarified this in the Discussion.

Comments 4: The annual incidence of cervical cancer is in single digits. It can be traced whether the cervical cancer patients have not been vaccinated or have not received catch-up vaccination, which can better prove the preventive effect of vaccination on CIN3+ cases and cervical cancer patients.

Response 4: We agree. In this study we were limited to an anonymized dataset approved by the Regional Ethics Committee, which did not allow individual linkage to the national vaccination registry (SYSVAK) or to HPV genotyping. We therefore could not document, for each cancer case, whether the woman was unvaccinated or vaccinated late. However, unpublished data from the Norwegian Cancer Registry indicate that the few cervical cancer cases occurring in vaccinated birth cohorts (born 1997 and later) have almost exclusively been in women who were not vaccinated, and that no cancers in women vaccinated before 15 years of age were caused by HPV16/18. In the catch-up cohorts (born 1991–1996), where vaccination was given in the 20s, the cancers that do occur are consistent with infections acquired before vaccination. We have added this explanation to the Discussion as context and retained the lack of individual linkage as a limitation.

Comments 5: This study only investigated the HPV vaccination status and CIN3+ cases of 25-29-year-old women. The incidence of CIN3+ cases or cervical cancer and its relationship with non-vaccination among women aged 29 and above until death should also be investigated.

Response 5: This study was deliberately restricted to women aged 25–29 years, i.e. the age of entry to organized screening and the first birth cohorts in Norway with substantial HPV vaccination coverage. Older women (≥30 years) are largely unvaccinated in Norway, and CIN3+/cervical cancer in those age groups is already being monitored at national level through linkage between the Cancer Registry, the Cervical Cancer Screening Programme and SYSVAK. Extending our regional analysis to older age groups would require individual-level linkage and a longer follow-up window to capture cancers occurring later in life. We have noted this as an avenue for future research.

Comments 6: Since 2010, the 12-year-old girls who received HPV vaccination in 1997 have entered the lower limit of the monitoring age of 25. If CIN3+ cases or cervical cancer patients occur among the vaccinated girls monitored from this time, it may be due to HPV type mismatch, which also verifies the efficacy of the HPV vaccine from another aspect.

Response 6: We agree. From 2022 onward, the first fully school-vaccinated cohort (born 1997, vaccinated at 12 years) entered screening at age 25, and any CIN3+ or cancer detected in such women is likely to represent infection with non-vaccine HPV types or very unusual vaccine-type breakthrough. In our regional data, numbers are too small to analyze this separately, and our anonymized dataset did not include HPV genotyping. National monitoring in Norway, where cervical lesions from vaccinated cohorts are HPV-typed and linked to SYSVAK, has so far shown a marked reduction in HPV16/18 and a higher proportion of non-16/18 types in lesions from vaccinated women, which is consistent with high vaccine effectiveness against the targeted types. We have added this point to the Discussion as supporting context.

Comments 7: The "1,2" after the author Elin Synnøve Mortensen should be superscript.

Response 7: Thank you for noting this formatting error. The affiliation numbers after Elin Synnøve Mortensen are now in superscript.

Round 2

Reviewer 2 Report

Comments and Suggestions for Authors

Thank you to the authors for the substantial revision of their very interesting manuscript. All the comments were addressed satisfactorily.